

# The Role of Organic Acids in New Particle Formation from Methanesulfonic Acid and Methylamine

Rongjie Zhang[1#], Jiewen Shen[1#], Hong-Bin Xie[1*], Jingwen Chen[1] and Jonas Elm[2]

[1]Key Laboratory of Industrial Ecology and Environmental Engineering (Ministry of Education), School of Environmental
Science and Technology, Dalian University of Technology, Dalian 116024, China
[2]Department of Chemistry and iClimate, Aarhus University, Langelandsgade 140, DK-8000 Aarhus C, Denmark

*Correspondence to*: Hong-Bin Xie (hbxie@dlut.edu.cn)

**Abstract.** Atmospheric organic acids (OAs) are expected to enhance methanesulfonic acid (MSA)-driven new particle formation (NPF). However, the exact role of OAs in MSA-driven NPF remains unclear. Here, we employed a two-step strategy to probe the role of OAs in MSA-methylamine (MA) NPF. Initially, we evaluated the enhancing potential of 12 commonly detected OAs in ternary MA-MSA-OA cluster formation by considering the formation free energies of the $(MSA)_1(MA)_1(OA)_1$ clusters and the atmospheric concentrations of the OAs. It was found that formic acid (ForA) has the highest potential to stabilize the MA-MSA clusters. The high enhancing potential of ForA results from its acidity, structural factors such as no intramolecular H-bonds and high atmospheric abundance. The second step is to extend the MSA-MA-ForA system to larger cluster sizes. The results indicate that ForA can indeed enhance MSA-MA NPF at atmospheric conditions (the upper limited temperature is 258.15 K), indicating that ForA might have an important role in MSA-driven NPF. The enhancing effect of ForA is mainly caused by an increased formation of the $(MSA)_2(MA)_1$ cluster, which is involved in the pathway of binary MSA-MA nucleation. Hence, our results indicate that OAs might be required to facilitate MSA-driven NPF in the atmosphere.

## 1 Introduction

New particle formation (NPF) accounts for a substantial fraction of atmospheric aerosols (Elm et al., 2020; Zhang et al., 2012; Lee et al., 2019; Yin et al., 2021), which exert significant effects on air quality, human health, and the global climate (Ho et al., 2007; Heal et al., 2012; Zhang et al., 2015; An et al., 2016). Evidences from field observations, laboratory experiments and simulations suggest that sulfuric acid (SA) is a key species in driving NPF over land (Lee et al., 2019; Elm et al., 2020; Yao et al., 2018; Sipila et al., 2010; Kirkby et al., 2011; Almeida et al., 2013; Nieminen et al., 2009). Recently, methanesulfonic acid (MSA)-driven NPF has received increasing attention with the implementation of stricter regulations on anthropogenic $SO_2$ emissions globally (Dawson et al., 2012; Chen et al., 2015; Arquero et al., 2017a; Arquero et al., 2017b; Chen and Finlayson-Pitts, 2017; Xu et al., 2017; Wen et al., 2018; Chen et al., 2020b; Perraud et al., 2020b; Perraud et al.,





2020a; Stahl et al., 2020), since MSA has a considerable contribution to NPF under the scenario that only natural sources of
$SO_2$ are kept (Perraud et al., 2015).

      Atmospheric bases can enhance both SA-driven and MSA-driven NPF by stabilizing freshly formed molecular clusters via acid-base proton transfer reactions (Zhang et al., 2012; Almeida et al., 2013; Chen et al., 2020b; Perraud et al., 2020b; Perraud et al., 2020a; Kurten et al., 2008; Loukonen et al., 2010; Glasoe et al., 2015; Olenius et al., 2017; Ma et al., 2019b; Zuo et al., 2021; Shen et al., 2020; Shen et al., 2019; Chen et al., 2020a; Xu et al., 2020; Nishino et al., 2014). Amines have
been shown to be much more efficient stabilizing species compared to ammonia for both SA-base and MSA-base cluster formation (Almeida et al., 2013; Kurten et al., 2008; Chen et al., 2020b; Perraud et al., 2020a; Shen et al., 2020; Chen et al., 2016). However, even considering the enhancing effect of bases, the SA-driven and MSA-driven nucleation mechanism still underestimates the nucleation rates compared to field observations (Zhang et al., 2012; Lee et al., 2019). This motivates the investigation of nucleation mechanisms involving other atmospheric relevant precursor vapors.

40          Atmospheric organic acids (OAs), originating from biogenic sources (vegetation) (Fehsenfeld et al., 1992; Chebbi and Carlier, 1996; Kavouras et al., 1998), anthropogenic sources (biomass burning and fossil fuel combustion) (Chebbi and Carlier, 1996; Chan et al., 2005; Falkovich et al., 2005; Cong et al., 2015; Friedman et al., 2017), and photochemical transformations from precursors might be relevant for NPF (Zhang et al., 2004; Zhang et al., 2012; Lee et al., 2019; Kavouras et al., 1998; Chebbi and Carlier, 1996; Paulot et al., 2009; Carlton et al., 2006). OAs cover a broad range of
species in the atmosphere with an estimated total atmospheric concentration above parts per billion (ppb) level (Chebbi and Carlier, 1996; Veres et al., 2011). Although nucleation from OAs alone leads to a limited contribution to NPF (Zhang et al., 2012), previous studies have indicated that OAs might enhance SA-driven nucleation via hydrogen-bond (H-bond) interactions between OAs and SA (Zhang et al., 2004; Zhang et al., 2012; Elm et al., 2020; Wang et al., 2019; Shi et al., 2019). Furthermore, OAs such as lactic-, glyoxylic-, glycolic- and malonic acid have been identified to enhance binary SA-
base nucleation, especially at very low temperatures (218 K) (Zhang et al., 2017; Li et al., 2017; Liu et al., 2018; Zhang et al., 2018). Under certain conditions, the ternary SA-base-OA system might present higher nucleation rates compared to the binary systems of SA-base, SA-OA and OA-base. In contrast to SA-driven NPF, MSA-base cluster formation is not very efficient on its own and thereby could be highly dependent on the enhancing effect of other contributing vapors (Dawson et al., 2012; Elm, 2021). While the SA-driven nucleation involving OAs has been extensively studied (Zhang et al., 2004; Elm
et al., 2017; Li et al., 2017; Zhang et al., 2017; Liu et al., 2018; Zhang et al., 2018; Wang et al., 2019; Shi et al., 2019), little is still known about the corresponding MSA-driven nucleation mechanism.

      The potentially important role of OAs in MSA-driven nucleation has been pointed out by previous studies, which focused on the interaction between MSA and several organic acid monomers or role of oxalic acid in MSA-amine NPF (Zhao et al., 2017; Arquero et al., 2017a; Arquero et al., 2017b; Xu et al., 2017; Sheng et al., 2019). However, to date, the
underlying enhancing effects from a broad range of OAs on MSA-amine nucleation have not been examined. Based on our recent work, the enhancing potential of other precursors on MSA-driven nucleation is quite sensitive to the specific



molecular structure (Shen et al., 2020; Shen et al., 2019). In addition, there is a large difference in atmospheric abundance of various OAs which also has an influence on the relative importance of different OAs in MSA-amine-OA cluster formation. Therefore, it is desirable to evaluate the role of OAs in MSA-driven NPF by considering the interaction of the different OAs with MSA-amine clusters and the atmospheric abundance of OAs.

Among the most commonly detected alkyl amines (methylamine (MA), dimethylamine (DMA) and trimethylamine (TMA)), MA has been identified to be the strongest enhancing agent in MSA-driven NPF (Chen et al., 2016). Hence, in this study we will focus on the enhancing effect of OAs on binary MSA-MA nucleation. We initially screen the potential role of 12 commonly detected atmospheric OAs (formic (ForA), $CH_2O_2$; acetic (AceA), $C_2H_4O_2$; glyoxylic (GlyA), $C_2H_2O_3$; oxalic (OxaA), $C_2H_2O_4$; pyruvic (PyrA), $C_3H_4O_3$; malic (MalA), $C_4H_6O_5$; maleic (MaleA), $C_4H_4O_4$; succinic (SucA), $C_4H_6O_4$; glutaric (GluA), $C_5H_8O_4$; adipic (AdiA), $C_6H_{10}O_4$; benzoic (BenA), $C_7H_6O_2$; and pinonic (PinA), $C_{10}H_{16}O_3$ acids) in ternary MA-MSA-OA nucleation by examining the formation of the $(MSA)_1(MA)_1(OA)_1$ clusters. Based on the calculated formation free energy ($\Delta G$) of the $(MSA)_1(MA)_1(OA)_1$ clusters and reported atmospheric concentrations of OAs, we rank the capability of different OAs to participate in ternary MSA-MA-OA nucleation. The OA with the highest capability was selected as a representative to further probe the enhancing potential of OAs on the MSA-MA NPF by considering the larger $(MSA)_x(MA)_y(OA)_z$ ($0 \leq y \leq x+z \leq 3$) clusters.

## 2 Computational Details

### 2.1 Configurational Sampling

A multi-step sampling scheme was used to search for the global minima structures of the 12 $(MA)_1(MSA)_1(OA)_1$ clusters, as well as the larger $(MSA)_x(MA)_y(OA)_z$ ($0 \leq y \leq x+z \leq 3$) clusters. The binary MSA-MA cluster structures were taken from our previous work (Shen et al., 2020). Details of the scheme can be found in our previous studies addressing atmospheric cluster formation (Shen et al., 2019; Shen et al., 2020; Xie et al., 2017; Ma et al., 2019a). Briefly, around 1000-10000 initial configurations for each cluster were randomly generated, and subsequently underwent a gradual screening process using various theoretical methods, to locate the clusters lowest in free energy. The employed theoretical methods for geometry optimization and single-point energy calculations include PM6, ωB97X-D/6-31+G(d,p), ωB97X-D/6-31++G(d,p), and DLPNO-CCSD(T)/aug-cc-pVTZ. The combination of ωB97X-D/6-31++G(d,p) and DLPNO-CCSD(T)/aug-cc-pVTZ has in several benchmarks shown good performance for studying the formation of atmospheric molecular clusters (Elm et al., 2013; Elm and Kristensen, 2017). All calculations employing semiempirical (PM6) and density functional theory (ωB97X-D functional) methods were performed in the GAUSSIAN 09 program package (Frisch et al., 2009), and DLPNO-CCSD(T)/aug-cc-pVTZ calculations were performed using the ORCA 4.0.0 program (Neese, 2012). The binding free energy ($\Delta G$) for each cluster was obtained by subtracting the Gibbs free energy of the constituent monomers (MA, MSA, and OA)



from that of the cluster at 298.15 K. The $\Delta G$ values at other temperatures were also calculated under the assumption that the enthalpy and entropy change ($\Delta H$ and $\Delta S$) are constant in the considered tropospheric temperature range (238.15-298.15 K).

## 2.2 Estimating the Concentrations of the $(MSA)_1(MA)_1(OA)_1$ Clusters

The atmospheric relevance of the studied $(MSA)_1(MA)_1(OA)_1$ clusters is not only determined by the thermodynamics of cluster formation, but also by the atmospheric abundance of the precursor molecules. The concentrations of the $(MSA)_1(MA)_1(OA)_1$ clusters were estimated according to the law of mass balance by the following equation:

$$[(MSA)_1(MA)_1(OA)_1]=[MSA]\times[MA]\times[OA]\times e^{\left(\frac{-\Delta G}{RT}\right)} \tag{1}$$

where $[(MSA)_1(MA)_1(OA)_1]$, $[MSA]$, $[MA]$ and $[OA]$ are the atmospheric concentrations of the $(MSA)_1(MA)_1(OA)_1$ cluster,
MSA, MA, and OA, respectively. $\Delta G$ is the calculated binding free energy, R is the molar gas constant and $T$ is the temperature (298.15 K).

The concentrations of precursors used in this study are based on reported values from field campaigns (see Table S1). Notably, except for ForA and AceA, the concentrations of the OAs were estimated from those measured in the particle phase. Hence, the values in Table S1 may overestimate the gaseous concentrations of the remaining 10 OAs and should be
considered as an upper limit. The concentration of the $(MSA)_2(MA)_1$ cluster was also calculated as a comparison. $[MA]$ and $[MSA]$ were set to be 10 ppt and $10^7$ molecules cm$^{-3}$ in the calculations, respectively. It should be noted that the choice of $[MA]$ and $[MSA]$ does not influence the calculated ranking of the OAs.

## 2.3 ACDC Simulations

        ACDC was employed to study the time evolution of cluster formation rates, steady-state cluster concentrations, and
growth pathways for the larger MSA-MA-OA cluster system (Mcgrath et al., 2012). We refer to our previous studies for further details of the theory behind ACDC (Shen et al., 2020; Shen et al., 2019). Here, the simulated system can be described as a "3×3 box" containing $(MSA)_x(MA)_y(OA)_z$ ($0 \leq y \leq x+z \leq 3$) clusters, where 3 represents the maximal number of bases (MA) and acids (MSA and OA). We do not consider the MSA-MA-OA clusters that contain more bases than acids, as it has been shown that these usually have high evaporation rates and thereby are of little importance in NPF (Olenius et al., 2013).
The selection of boundary clusters was based on the stability of clusters under the studied temperature (see details in the Supplement). As there have not been reported any coagulation coefficient of MSA from field campaigns, we set the coagulation sink to a constant value of $2 \times 10^{-2}$ s$^{-1}$, corresponding to a typical value in polluted regions (Yao et al., 2018). In addition, the temperature and concentrations of precursor gases (MSA, MA, and OA) were varied in the simulations to probe the effect of various atmospheric conditions. The simulations were performed at 238.15-298.15 K to investigate the effect of
temperature. The test ranges for the OA precursor concentrations are $10^8$-$10^{12}$ cm$^{-3}$. $[MA]$ and $[MSA]$ were set to be 1-100 ppt and $10^5$-$10^8$ cm$^{-3}$, respectively.





# 3 Results and Discussion

## 3.1 Analysis of the $(MSA)_1(MA)_1(OA)_1$ Clusters

The obtained global minima of 12 $(MSA)_1(MA)_1(OA)_1$ clusters are presented in Fig. 1. For all $(MSA)_1(MA)_1(OA)_1$
clusters, proton transfer from MSA to MA occurs, and the clusters are stabilized via both intermolecular H-bonds and
electrostatic interactions between positive and negative species. It should be noted that no proton transfer occurs from MSA
to MA in the binary $(MSA)_1(MA)_1$ cluster (Shen et al., 2020). Thus, the addition of OAs to the $(MSA)_1(MA)_1$ cluster
promotes the proton transfer. The formation of charged species could potentially enhance particle formation for ternary
MSA-MA-OA nucleation compared to the binary MSA-MA system.

The clusters, forming from all monocarboxylic acid except PinA, present almost identical intermolecular interaction
pattern. Four H-bonds are formed in these clusters and monomers are linked by two ten-membered rings and a six-membered
ring. PinA differs from the other monocarboxylic acids as the additional carbonyl group (C=O) is involved in the H-bond
formation in the $(MSA)_1(MA)_1(PinA)_1$ cluster. For clusters, forming from dicarboxylic acids, only MaleA and GluA can
interact with $(MSA)_1(MA)_1$ cluster via both -COOH groups in the $(MA)_1(MSA)_1(OA)_1$ clusters. However, other dicarboxylic
acids follow similar intermolecular interaction pattern as ones containing monocarboxylic acids. It deserves mentioning that
a significant deformation occurs in dicarboxylic acids when they interact with $(MSA)_1(MA)_1$ cluster via both -COOH,
compared to their corresponding isolated conformation. A previous study also found that dicarboxylic acids can interact with
SA via both –COOH groups for sulfuric acid-OA dimer clusters (Elm et al., 2017).



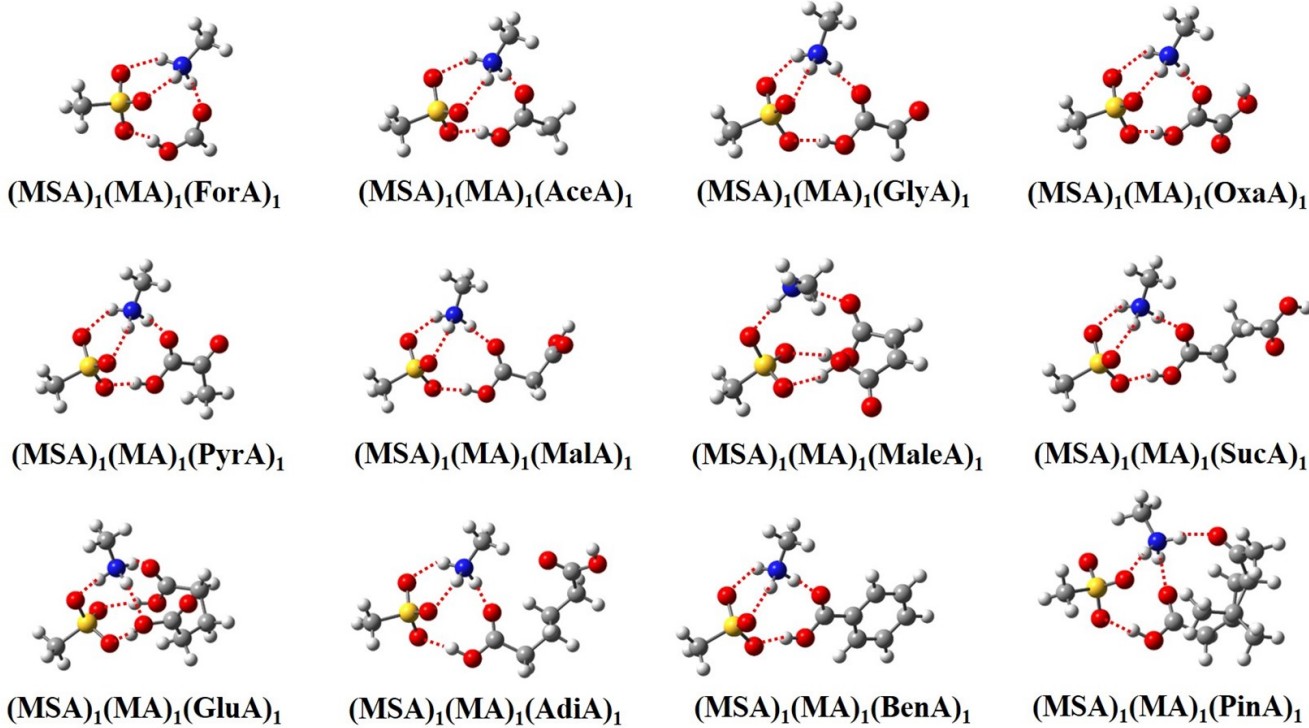

**Figure 1: Lowest Gibbs free energy conformations of 12 (MSA)₁(MA)₁(OA)₁ clusters (OA = ForA, AceA, GlyA, OxaA, PyrA, MalA, MaleA, SucA, GluA, AdiA, BenA, and PinA) calculated at the ωB97X-D/6-31++G(d,p) level of theory, at 298.15 K and 1 atm. The red balls represent oxygen atoms, blue = nitrogen atoms, gray = carbon atoms, and white = hydrogen atoms. The red dashed lines indicate hydrogen-bonds.**

Fig. 2a presents the $\Delta G$ values for 12 (MSA)₁(MA)₁(OA)₁ clusters, along with the calculated value for the (MSA)₂(MA)₁ cluster as a comparison. Generally, the $\Delta G$ values of the (MSA)₁(MA)₁(OA)₁ clusters vary from -12.69 to -17.87 kcal mol⁻¹, and are 2.49-7.67 kcal mol⁻¹ higher than that of the (MSA)₂(MA)₁ cluster. Based on previous studies, several factors such as the acidity and intramolecular H-bond of OAs can affect their interaction energy with amine or SA (Elm et al., 2017). It should be noted that the structures of the simple monocarboxylic acids ForA, AceA, and BenA almost do not change during the interaction with MSA-MA cluster, presenting the intrinsical interacting pattern of the –COOH group with the MSA-MA cluster. Therefore, the MSA-MA-OA systems with the simple monocarboxylic acids were initially selected to analyze which factors that affect the interaction between OAs and MSA-MA cluster. As shown in Fig. 2b, the p$K_a$ of the OA presents a good linear relationship with the calculated $\Delta G$ values for the formation of the (MSA)₁(MA)₁(OA)₁ clusters for the simple monocarboxylic acids, indicating the role of OA acidity in determining the $\Delta G$ value. This is the first time to reveal such the linear relationship of p$K_a$ with $\Delta G$ for OAs with MSA(SA)-amine clusters. In the following part, the linear relationship of p$K_a$ with $\Delta G$ value of the (MSA)₁(MA)₁(OA)₁ was used as a baseline to discuss other factors that affect the $\Delta G$ values for the formation of the (MSA)₁(MA)₁(OA)₁ clusters.





As shown in Fig. 2b, the $\Delta G$ values of GluA and PinA lie under the baseline, indicating there are favorable structural factors besides acidity to enhance the interaction of these two OAs with MSA-MA cluster. It was found that GluA and PinA form additional intermolecular H-bonds with MSA-MA cluster via additional -COOH and C=O, respectively. The formation of additional intermolecular hydrogen bonds explains why the $\Delta G$ values of GluA and PinA are lower than those predicted by their p$K_a$ (under the baseline).

The $\Delta G$ values of OxaA, PyrA, MaleA, GlyA, MalA, SucA and AdiA lie above the baseline, indicating there are unfavorable structural factors to restrain the interaction of these seven OAs with the MSA-MA clusters. By comparing the structure of OAs in the $(MSA)_1(MA)_1(OA)_1$ cluster and their corresponding isolated conformation (Fig. S1), it was found that the original intramolecular H-bonds of GlyA, OxaA, MaleA and PyrA have been broken in the interaction with the MSA-MA cluster. Therefore, an energetic penalty should be paid for breaking the original intramolecular H-bonds for these four acids, which should be the main reason that the $\Delta G$ values are higher than those evaluated by their p$K_a$ values (above the baseline). A similar phenomenon has previously been observed for $\alpha$-keto carboxylic acids in sulfuric acid-OA dimer clusters (Elm et al., 2017). The other three outliers (the dicarboxylic acids MalA, SucA and AdiA) interact with MSA-MA only via one –COOH group, behaving like a simple monocarboxylic acid. It was found that the structure of SucA and AdiA changes slightly during the interaction with MSA-MA cluster compared to their corresponding isolated conformations. The calculated root-mean-square-deviation (RMSD) of the OA structure in MSA-MA-OA cluster relative to their corresponding isolated conformers are 0.046 Å and 0.353 Å for SucA and AdiA, respectively. Therefore, the structural change upon cluster formation cannot explain the outliers. It was speculated that the clustering ability of the dicarboxylic acids follow an alternating even/odd pattern, as recently observed for carboxylic acid dimer clusters (Elm et al., 2019). The structure of MalA changes significantly during the interaction with the MSA-MA cluster with a RMSD of 1.27 Å, which explains the reason for behaving as an outlier. Overall, our results show that the acidity, hydrogen-bond forming capacity, structural deformation energy of OA and even/odd pattern if it is dicarboxylic acid are the main factors determining the formation free energy of MSA-MA-OA cluster.





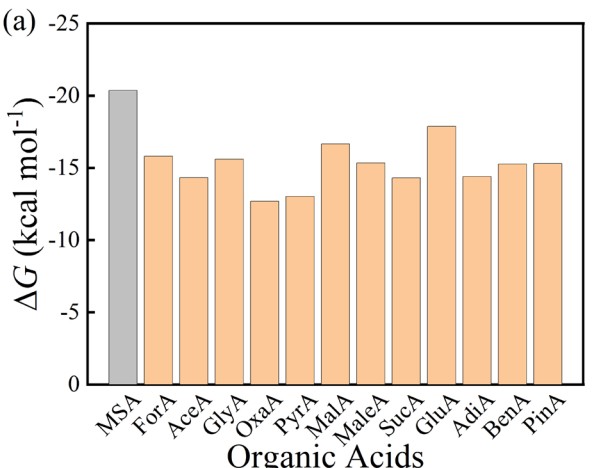

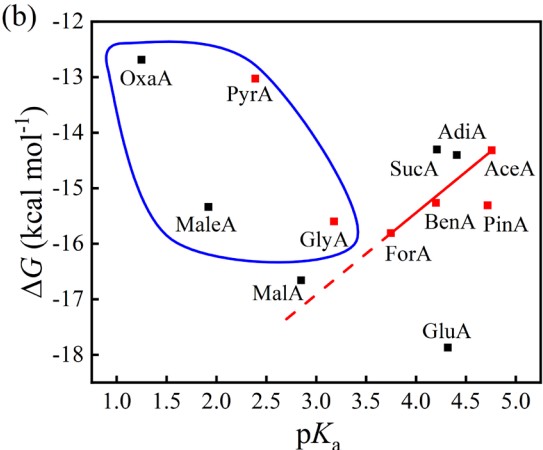


**Figure 2: Binding free energy ($\Delta G$) (kcal mol$^{-1}$) of the 12 (MSA)$_1$(MA)$_1$(OA)$_1$ clusters and the (MSA)$_2$(MA)$_1$ cluster at the DLPNO-CCSD(T)/aug-cc-pVTZ//ωB97X-D/6-31++G(d,p) level of theory (the calculations are performed at 298.15 K and 1 atm) (a), and the relationship between the p$K_a$ of the organic acids with the binding free energy ($\Delta G$) (kcal mol$^{-1}$) of the 12 (MSA)$_1$(MA)$_1$(OA)$_1$ clusters at the DLPNO-CCSD(T)/aug-cc-pVTZ//ωB97X-D/6-31++G(d,p) level of theory (the calculations are performed at 298.15**

**K and 1 atm) (b). The solid line in (b) is the fitted linear regression line of the $\Delta G$ on p$K_a$ value for the simplest monocarboxylic acids with $r$ = 0.996, and the dashed line is the extension of the solid line. The red squares represent monocarboxylic acids, the black squares represent dicarboxylic acids, and the circled species contain intramolecular H-bond.**

To evaluate the potential of the 12 OAs in ternary MA-MSA-OA nucleation, the atmospheric concentrations of (MSA)$_1$(MA)$_1$(OA)$_1$ were calculated by Eq.(1). Fig. 3 depicts the calculated [(MSA)$_1$(MA)$_1$(OA)$_1$] along with

[(MSA)$_2$(MA)$_1$]. As can be seen from Fig. 3, the [(MSA)$_1$(MA)$_1$(OA)$_1$] span many orders of magnitude (from ~10$^{-7}$ to ~1 molecules cm$^{-3}$). Based on the mean values, [(MSA)$_1$(MA)$_1$(OA)$_1$] decreases in the order: ForA > AceA ≈ MSA> GluA > MalA > BenA > GlyA > PinA > SucA > MaleA > OxaA > AdiA > PyrA. Therefore, ForA should have the highest potential for participating in MSA-MA-OA cluster formation among the 12 considered OAs. More importantly, the calculated [(MSA)$_1$(MA)$_1$(ForA)$_1$] is higher than [(MSA)$_2$(MA)$_1$], implying ForA could participate in MSA-MA nucleation by partly

replacing MSA. In the following, larger MSA-MA-ForA clusters were investigated to further probe the possible enhancing potential of ForA on MSA-MA nucleation.

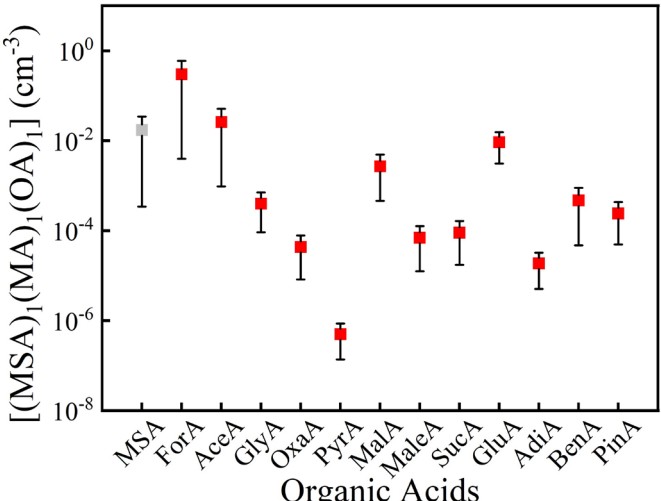

**Figure 3: The calculated [(MSA)$_1$(MA)$_1$(OA)$_1$] and [(MSA)$_2$(MA)$_1$] (molecules cm$^{-3}$) based on the mass balance equation and reported concentrations of precursors. The bars denote the minimum and the maximum of [(MSA)$_1$(MA)$_1$(OA)$_1$] and [(MSA)$_2$(MA)$_1$], and the red square denotes the mean values.**

### 3.2 Analysis of larger MSA-MA-ForA Clusters

The pure (MSA)$_{1-3}$ and binary (MSA)$_x$(MA)$_y$ ($1 \leq y \leq x \leq 3$) clusters have been discussed in our previous study (Shen et al., 2020; Shen et al., 2019). Here we mainly focus on the ForA-containing clusters. Fig. S2 presents the obtained global minimum structures of the ForA-containing clusters. Similar to the pure (MSA)$_{1-3}$ clusters, there is observed no proton transfer in the pure acid clusters MSA-ForA and (ForA)$_{1-3}$. For the binary MA-ForA clusters, proton transfer does not occur in the (MA)$_1$(ForA)$_{1-2}$ clusters and a single MA is protonated in the (MA)$_1$(ForA)$_3$ and (MA)$_2$(ForA)$_2$ clusters. In the (MA)$_2$(ForA)$_3$ and (MA)$_3$(ForA)$_3$ clusters we observe two and three proton transfers, respectively. For the ternary MSA-MA-ForA cluster systems, all MSA molecules have transferred a proton to MA and ForA have transferred a proton to the remaining MA if they have when $x \leq y$. When $x > y$ only some of the MSA have transferred a proton to MA and ForA are kept intact. Therefore, proton transfer from MSA to MA is more favorable than that from ForA to MA, which is in accordance with the lower acidity of ForA compared to MSA.

Fig. S3 shows the calculated $\Delta G$ values of the (MSA)$_x$(MA)$_y$(ForA)$_z$ ($0 \leq y \leq x+z \leq 3$) clusters. Generally, the more ForA monomers present in the cluster the higher the $\Delta G$ value is when the number of total acids and ratio of acid to MA are equal. Based on calculated $\Delta G$ values of all considered clusters, the evaporation rates of the clusters were calculated. A previous study established that it is important to consider the concentration of precursor when evaluating the cluster stability (Xie et al., 2017). Generally, given higher concentration of precursors, the collision probability for a cluster is increased and the balance between collision and evaporation is shifted forward, resulting in a higher cluster stability. Since the atmospheric concentration of ForA is much higher (about $10^3$) than that of MA and MSA, an effective evaporation rate was used to





evaluate cluster stability of ForA-containing clusters. The effective evaporation rates of the ForA-containing clusters were

roughly obtained by scaling the original evaporation rate by $10^{-3}$, due to the concentration difference. The evaporation rates of binary MSA-MA clusters and effective ones of ForA-containing clusters are presented in Fig. 4. Among the ForA-containing clusters, only the evaporation rates of the $(MSA)_1(MA)_1(ForA)_1$ and $(MSA)_1(ForA)_1$ clusters are lower than those of corresponding binary MSA-MA clusters with equal number of acids and bases. The $(ForA)_2$ cluster is found to be comparable to $(MSA)_2$. The remaining ForA-containing clusters have much higher evaporation rates compared to the

corresponding binary MSA-MA clusters. Therefore, the $(MSA)_1(MA)_1(ForA)_1$ and $(MSA)_1(ForA)_1$ clusters have the highest potential to participate in MSA-MA nucleation, followed by $(ForA)_2$. It deserves mentioning that the evaporation rates of all ForA-containing clusters are higher than those of corresponding binary MSA-MA clusters if the concentration difference of precursors was not considered. Therefore, it is the high concentration of ForA that drives the stability of the clusters and not the intrinsic evaporation rate.

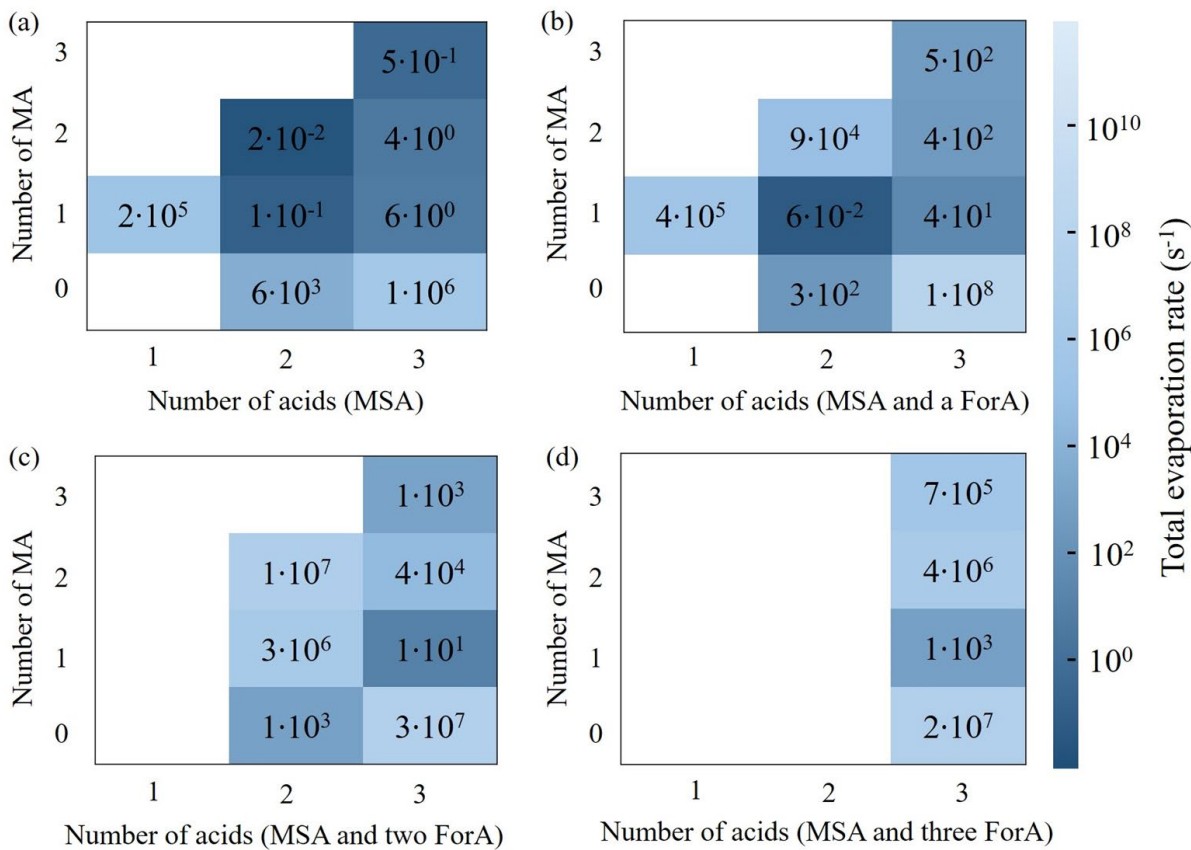

**Figure 4: Calculated effective evaporation rates (s⁻¹) of the $(MSA)_x(MA)_y(ForA)_z$ ($0 \leq y \leq x+z \leq 3$) clusters at 278.15 K. a) without ForA monomer, b) containing 1 ForA monomer, c) containing 2 ForA monomers, and d) containing 3 ForA monomers.**





### 3.3 Enhancing Effect of ForA

Here, we employed the enhancing coefficient $R$, as the ratio of cluster formation rate ($J$) for the ternary MSA-MA-ForA

cluster system ($J_{MA-MSA-ForA}$) relative to that of the binary MSA-MA cluster system ($J_{MA-MSA}$) to present the enhancing effect

of ForA. For $R$ above 1, it can be inferred that ForA has an enhancing effect on the binary MSA-MA system. Previous

studies have shown that the temperature and concentrations of precursors have a large influence on $R$ of OAs for the ternary

SA-base-OA systems (Elm et al., 2020; Zhang et al., 2017; Li et al., 2017; Liu et al., 2018; Zhang et al., 2018). Herein, the

effects of temperature and precursor concentration on $R$ were examined for the ternary MSA-MA-ForA cluster system. Two

steps were employed to investigate the effects of temperature and precursor concentration on the $R$. First step investigated

the variation of $R$ with [ForA] ($10^8$-$10^{12}$ cm$^{-3}$) at different temperature (238.15-298.15 K). The second step investigated the

variation of $R$ with [MA] (1-100 ppt) or [MSA] ($10^5$-$10^8$ cm$^{-3}$) at fixed temperature and [ForA].

The variation of $R$ with [ForA] at 238.15, 258.15, 278.15 and 298.15 K, [MA]=10 ppt and [MSA] =$10^7$ molecules cm$^{-3}$

is presented in Fig. 5a. As seen from Fig. 5a, $R$ almost remains as unity at all considered temperature range when [ForA] is

less than $10^9$ molecules cm$^{-3}$. With increasing [ForA] when [ForA] is more than $10^9$ molecules cm$^{-3}$, $R$ starts to obviously

increase with [ForA] at low temperature (< 258.15 K). However, the effect of [ForA] on $R$ is much less pronounced when

temperature is higher than 278.15 K, and the increase of $R$ becomes nonnegligible only at 278.15 K and high [ForA] (>$10^{11}$

cm$^{-3}$). Therefore, both temperature and [ForA] can significantly affect $R$. Similar results were also reported in previous

studies on effect of OAs on SA-NH$_3$ nucleation (Zhang et al., 2017; Liu et al., 2018; Zhang et al., 2018). It deserves

mentioning that the upper limited temperature that ForA can effectively enhance MSA-MA NPF (258.15K) is about 40 K

higher than those of the previously studied OAs enhancing SA-NH$_3$ NPF. In the following part, the condition with a

reachable temperature ($T$ = 258.15 K) and [ForA] ($10^{11}$ cm$^{-3}$ (~ 4 ppb)) in the ambient atmosphere (Khwaja, 1995), was

employed to investigate the effect of [MA] and [MSA] on $R$. It should be mentioned that these are not unrealistic conditions

for instance at polluted sites in inland China.

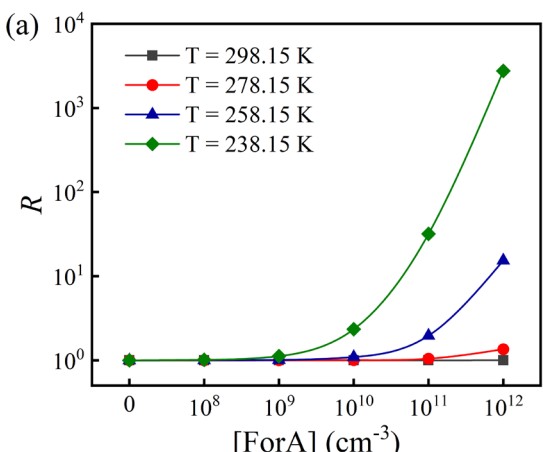
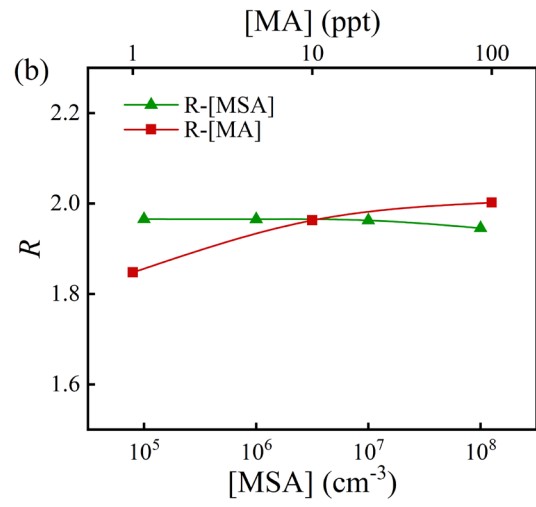




**Figure 5: The enhancing coefficient ($R$) as a function of [ForA] at [MA] = 10 ppt, [MSA] = $10^7$ cm$^{-3}$, and four temperatures (238.15, 258.15, 278.15, and 298.15 K) (a), and as a function of [MSA]/[MA] at [MA]=10 ppt/[MSA] = $10^7$ cm$^{-3}$, [ForA] = $10^{11}$ cm$^{-3}$ and T = 258.15 K (b).**

As can be seen in Fig. 5b, $R$ shows a small but positive dependence on [MA]. However, $R$ has little correlation with [MSA]. The difference between the relationships of $R_{[MA]}$ and $R_{[MSA]}$ can be explained by the difference in cluster stability and relative abundance of the precursors in the system. At [ForA] = $10^{11}$ molecules cm$^{-3}$, the abundance of precursors in the system approximately decreases in the order ForA > MA > MSA. Due to the scarcity of MSA in the system, increased MA will prefer to participate in the formation of ForA-containing clusters. Therefore, increased [MA] has a small positive effect on $R$. Additionally, since the binding potential of MSA with the MA is much higher than that with MA-ForA clusters, the increased number of MSA molecules will primarily cluster with MA to form binary MSA-MA clusters and consequently the influence on the ternary MSA-MA-ForA pathways is minor.

**3.4 Enhancement Mechanism of ForA**

The enhancement mechanism of ForA can be disclosed by a comparative analysis on the cluster growth pathway of ternary MSA-MA-ForA and binary MSA-MA cluster system. Fig. 6a shows the main cluster growth pathways for the ternary MSA-MA-ForA cluster system at the conditions where ForA showed an enhancing effect ($T$ = 258.15 K, [MA] = 10 ppt, [MSA] = $10^7$ molecules cm$^{-3}$, and [ForA] = $10^{11}$ molecules cm$^{-3}$). The cluster growth pathways for the binary MSA-MA system at the same condition (except for [ForA] = 0 molecules cm$^{-3}$) are shown in Fig. 6b.

For binary MSA-MA nucleation, the cluster growth mainly proceeds via the following route: MSA → $(MSA)_2$ → $(MSA)_2(MA)_1$ → $(MSA)_2(MA)_2$ → $(MSA)_3(MA)_2$ → $(MSA)_3(MA)_3$, fluxing out of the system as the $(MSA)_4(MA)_3$ cluster, consistent with our previous findings.(Shen et al., 2020) In addition, the $(MSA)_3(MA)_1$ cluster also has a nonnegligible contribution (8%) to the formation of the $(MSA)_3(MA)_2$ cluster. As seen in Fig. 6a, when present at a concentration of $10^{11}$ molecules cm$^{-3}$, ForA initially participates in the formation of smaller binary MSA-MA clusters and subsequently evaporates from them via three routes,i.e. MSA → $(MSA)_1(ForA)_1$ → $(MSA)_2(ForA)_1$ → $(MSA)_2$; MSA → $(MSA)_1(ForA)_1$ → $(MSA)_1(ForA)_1(MA)_1$ → $(MSA)_2(ForA)_1(MA)_1$ → $(MSA)_2(MA)_1$ and MSA → $(MSA)_1(MA)_1$ → $(MSA)_1(ForA)_1(MA)_1$ →$(MSA)_2(ForA)_1(MA)_1$ → $(MSA)_2(MA)_1$. Therefore, ForA plays a catalytic role in the formation of the smaller binary MSA-MA clusters. In all ForA-containing clusters involved in the catalytic process, only $(MSA)_1(ForA)_1$ and $(MSA)_1(ForA)_1(MA)_1$ can continue to grow, consistent with their stability. As ForA-driven catalytic process is the only difference between formation pathway of the ternary MSA-MA-ForA and the binary MSA-MA cluster system, it should be the main reason that the presence of ForA can enhance the nucleation ($R$ = 1.96), compared to the case of binary MSA-MA system. In addition, ForA mainly contributes to the formation of $(MSA)_2(MA)_1$ clusters (51%) in the catalytic process. Therefore, ForA enhances the binary MSA-MA nucleation mainly via catalyzing the formation of the $(MSA)_2(MA)_1$ cluster. As can be seen from Fig. S4-5, the catalyzed formation of the $(MSA)_2(MA)_1$ cluster is the main route for the ForA enhancing binary MSA-MA nucleation when [MA] and [MSA] changes.



(a)

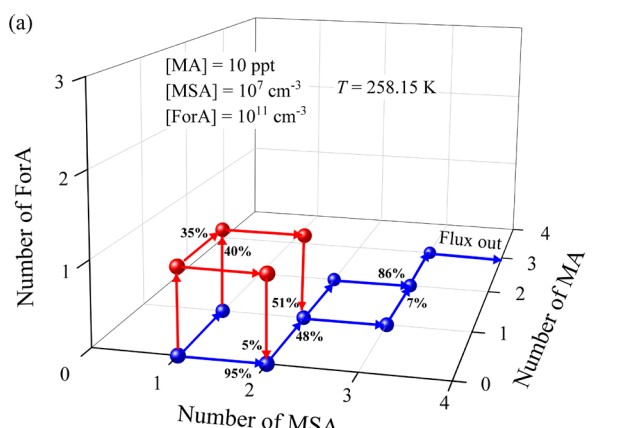

(b)

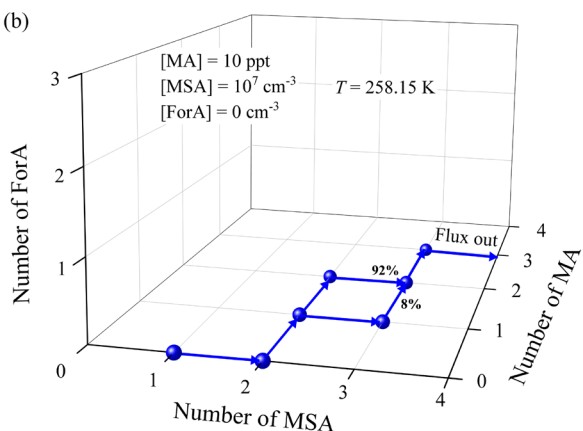

**Figure 6: Main cluster formation pathways for the ternary MSA-MA-ForA system (a) and the MSA-MA system (b) at $T$ = 258.15 K, [MA] = 10 ppt, [MSA] = $10^7$ molecules cm$^{-3}$, and [ForA] = $10^{11}$ molecules cm$^{-3}$ (only for the ternary MSA-MA-ForA system).**

### 3.5 Atmospheric Implications

In this study, the potential of 12 commonly detected OAs in MSA-MA-OA nucleation has been systematically evaluated by examining the formation of the $(MSA)_1(MA)_1(OA)_1$ clusters. It was found that ForA has the highest potential to participate in ternary MSA-MA-OA nucleation. This study also suggests that for a given OA to have highest potential in ternary MSA-MA-OA nucleation, the OA should have the following features: high atmospheric abundance, strong acidity, strong H-bond forming capacity, weak or no intramolecular interaction in the monomer, and no/little structural deformation during clustering. It is noted that besides the simple OAs studied here, there are some complex organics with multi-functional groups including -COOH, e.g. highly oxygenated molecules (HOMs), in the atmosphere. These complex organics also might participate in the MSA-MA nucleation since they could interact with the MSA-MA cluster in a similar fashion as the simple OAs (H-bonds and acid-base reaction). Therefore, studies on the role of the complex organics with multi-functional groups in MSA-MA NPF are warranted to further augment the current understanding of MSA-driven NPF.

This study for the first time reveals that ForA exerts a catalytic enhancing effect on binary MSA-MA nucleation by facilitating the formation of clusters in the initial stage of NPF. Based on reported energetic data of $(SA)_1(amine)_1(OAs)_1$ (amine = MA and DMA, OAs = ForA, AceA, OxaA, PyrA, MalA, MaleA, SucA, GluA, AdiA, BenA and PinA) and eq.(1) (Li et al., 2020), $(SA)_1(amine)_1(ForA)_1$ is calculated to have the (or second) highest concentration (see Table S3). Therefore, ForA could also have a high enhancing effect on SA-amine nucleation. A study on the enhancing effect of ForA on SA-driven NPF is warranted to further augment the current understanding of the contribution of OAs on NPF. The enhancing effect of ForA on binary MSA-MA nucleation is highly dependent on [ForA] and the temperature. At [ForA] =$10^{11}$ molecules cm$^{-3}$ and 258.15 K, a reachable concentration and temperature in ambient atmosphere, ForA can effectively enhance binary MSA-MA nucleation, clarifying its important role in MSA-driven NPF. In addition, the concentration of



ForA could be even higher in regions with significant primary emission sources and/or intense chemical production (Franco et al., 2021). Therefore, as a ubiquitous organic acid in the atmosphere, the contribution of ForA to NPF involving MSA and amines deserves more concerns in the future.


*Data availability.* The data in this article are available from the corresponding author upon request (hbxie@dlut.edu.cn).

*Author contribution.* HBX designed research; RJZ, JWS and HBX performed research; RJZ, JWS and HBX analyzed data; RJZ, JWS, HBX, JWC and JE wrote the paper; and HBX, JWC and JE reviewed and revised the paper.

*Competing interests.* The authors declare that they have no conflict of interest.

*Acknowledgements.* The study was supported by the National Natural Science Foundation of China (21876024), the Major International (Regional) Joint Research Project (21661142001) and Supercomputing Center of Dalian University of Technology. Jonas Elm thanks the Independent Research Fund Denmark grant number 9064-00001B.

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
