# Peer review of "The Role of Organic Acids in New Particle Formation from Methanesulfonic Acid and Methylamine"

_Atmospheric Chemistry and Physics, 2021_

## Author Comment (AC1)

**Responses to the Comments of the Reviewer**

**General comment**

**RC:** *The manuscript by Zhang et al. presents a theoretical evaluation on the role of organic acids in MSA-MA NPF. Among all considered organic acids, they found that formic acid (ForA) has the highest potential to stabilize the MA-MSA cluster and can enhance MSA-MA nucleation at atmospheric conditions via catalyzing the formation of clusters in the initial stage of NPF. The structural factors that affect the enhancing potential of organic acids on MSA-MA NPF were also revealed. The selected topic should be interesting across a range of atmospheric chemistry community. The work is technically well performed and calculated data can well support the conclusion. The manuscript is well written and easy to follow. Therefore, I recommend publication of this manuscript after consideration of the following comments.*

**AC:** Thanks for the positive comment. We have revised the manuscript to further enhance its quality.

**Special Suggestions and Comments**

**RC:** 1) *The accuracy of DLPNO-CCSD(T)/aug-cc-pVTZ calculation depends on selected keywords for the convergence. Please clearly present the selected keywords for the DLPNO-CCSD(T)/aug-cc-pVTZ calculation in the Computational Details section.*

**AC:** We agree with the reviewer that the exact keywords employed in the DLPNO-CCSD(T) calculations should be specified. Actually, we used tight PNO and SCF convergence criteria in the DLPNO-CCSD(T)/aug-cc-pVTZ single-point energy calculations by employing keywords tightpno and tightscf. In the revised manuscript, the following sentence was added:

"The tight PNO and SCF convergence criteria were used in DLPNO-CCSD(T)/aug-cc-pVTZ calculations by employing keywords tightpno and tightscf."

**RC:** 2) *Lines 82-83, in my opinion, the range for the number of initial configurations (1000-10000) for each cluster is large and it would be better to explain the reason.*

**AC:** Thanks for the suggestion. We agree with the reviewer that the range for the number of initial configurations (1000-10000) for each cluster is large. Exactly, we selected fewer initial configurations for clusters with fewer molecules and more initial configurations for clusters with more molecules since the configuration space for the cluster with more molecules is much larger than that for the cluster with fewer molecules. In the revised manuscript, the original sentence "Briefly, around 1000-10000 initial configurations for each cluster were randomly generated" was revised as "Briefly, around 1000-10000 initial configurations for each cluster (fewer initial configurations for clusters with fewer

molecules and more initial configurations for clusters with more molecules) were randomly generated".

**RC:** 3) *It is better to test how the selection of coagulation sink affects the core conclusion.*
**AC:** We appreciated the suggestion. We have tested the effect of coagulation sink coefficients on the enhancing coefficient $R$ and enhancement mechanism of ForA. The coagulation sink coefficients in the range from $2 \times 10^{-4}$ to $2 \times 10^{-2}$ s$^{-1}$, covering possible values in clean and haze days, were selected to test the effect. In the revised manuscript, the following sentences were added:

"In addition, the coagulation sink coefficients in the range from $2 \times 10^{-4}$ to $2 \times 10^{-2}$ s$^{-1}$, covering possible values in clean and haze days, were selected to test the effect of coagulation sink coefficients on the main results."

As shown in Figure S5, the selection of coagulation sink has no obvious effect on enhancing coefficient $R$. As shown in Figure S8, there is no significant difference for the cluster formation pathway when coagulation sink coefficient $2 \times 10^{-4}$ s$^{-1}$ and $2 \times 10^{-3}$ s$^{-1}$ were employed, indicating the selection of coagulation sink coefficient can't change the predicted catalysis mechanism of ForA. Therefore, the selection of coagulation sink coefficients does not affect our main conclusions. We added the Figure S5 and S8 in the revised Supplement. In the revised manuscript, the following sentences were added:

"As shown in Fig. S5, when coagulation sink coefficients change from $2 \times 10^{-4}$ s$^{-1}$ to $2 \times 10^{-2}$ s$^{-1}$, $R$ almost remains constant, indicating the selection of coagulation sink coefficient has no significant effect on $R$. This is not surprising, as $R$ represents the ratio of the new particle formation rates and thereby the effect of the choice of coagulation sink largely cancels out."

"In addition, as shown in Fig. S8, when coagulation sink coefficients $2 \times 10^{-4}$ s$^{-1}$ and $2 \times 10^{-3}$ s$^{-1}$ were employed, the formation pathway of the ternary MSA-MA-ForA clusters changes slightly compared to the case where a coagulation sink coefficient of $2 \times 10^{-2}$ s$^{-1}$ was used. Therefore, the selection of coagulation sink coefficient has no effect on the predicted catalysis mechanism of ForA."

[Figure]

**Figure S5. Variation of the enhancing coefficient ($R$) with coagulation sink coefficient (s$^{-1}$) at [MA] = 10 ppt, [MSA] = 10$^7$ cm$^{-3}$, [ForA] = 10$^{11}$ cm$^{-3}$ and T = 258.15 K.**

[Figure]

**Figure S8. Main cluster formation pathways for the ternary MA-MSA-ForA system at two different coagulation sink coefficients (2 × 10$^{-4}$ s$^{-1}$ (a) and 2 × 10$^{-3}$ s$^{-1}$ (b)), T = 258.15K, [MA] = 10 ppt, [MSA] = 10$^7$ cm$^{-3}$ and [ForA] = 10$^{11}$ cm$^{-3}$.**

**RC:** 4) *Line 98, please cite the references for the equation (1).*
**AC:** We certainly agree that this equation should be referenced. In the revised manuscript, the following reference was cited in the original sentence.
"Lin, Y., Ji, Y., Li, Y., Secrest, J., Xu, W., Xu, F., Wang, Y., An, T., and Zhang, R.: Interaction between succinic acid and sulfuric acid–base clusters, Atmos. Chem. Phys., 19, 8003-8019, http://doi.org/10.5194/acp-19-8003-2019, 2019."

**RC:** 5) *Lines 174-175, in order to explain why SucA and AdiA are outliers, the authors should clearly present the even/odd pattern of dicarboxylic acids.*
**AC:** Thanks for the suggestion. In the revised manuscript, the following sentences have been added to present the even/odd pattern of dicarboxylic acids. "The clustering ability of straight-chain dicarboxylic acids has been suggested to follow an alternating even/odd pattern by observing that the dimer formation for GluA (C5) is more efficient than SucA (C4) and AdiA (C6) (Elm et al., 2019). Therefore, it was speculated that the alternating even/odd pattern for the clustering ability of the dicarboxylic acids can explain SucA and AdiA as outliers."

**RC:** 6) *Please provide the equation for calculating the binding free energy of (SA)$_1$(amine)$_1$(OAs)$_1$ in Table S3 and point out whether the calculated [(SA)$_1$(amine)$_1$(OAs)$_1$] is the mean values based on concentration of precursors.*
**AC:** We appreciated the suggestion. The following equation for calculating the binding free energy of (SA)$_1$(amine)$_1$(OAs)$_1$ was provided in Table S3:

$$\Delta G = \Delta G_{R1} + \Delta G_{R2}$$

R1 presents the reaction SA + amine $\rightarrow$ (SA)$_1$(amine)$_1$ and R2 for reaction (SA)$_1$(amine)$_1$ + OA $\rightarrow$ (SA)$_1$(amine)$_1$(OA)$_1$.

We have specified that the calculated [(SA)$_1$(amine)$_1$(OAs)$_1$] are the mean values based on concentration of precursors. In the revised Supplement, the original table caption for Table S3 was changed to "The calculated mean concentrations of (SA)$_1$(amine)$_1$(OAs)$_1$ based on the mass balance equation, reported concentrations of precursors and energetic data of the (SA)$_1$(amine)$_1$(OAs)$_1$ clusters".

**RC:** 7) *Please check the guidelines of Atmos. Chem. Phys. for references, and all the references should be cited in the same style.*
**AC:** We have carefully checked the reference guidelines and made sure all the references were cited in the same style.

**RC:** 8) *Some minor mistakes are shown in the manuscript, e.g., Line 79, "global minimum structures" instead of "global minima structures"; Line 135, "ones" should be "clusters"; Lines 223-224, it should be written as " The evaporation rate of the (ForA)$_2$ cluster is found to be comparable to that of (MSA)$_2$".*
**AC:** We have corrected all these minor errors in the revised manuscript.

---

## Author Comment (AC2)

**Responses to the Comments of the Reviewer**

**General comment**

**RC:** *The manuscript by Zhang et al. is devoted to the role that organic acids (OA) may play in new particle formation from methanesulfonic acid (MAS) and methylamine (MA)molecules. It is based on high-level quantum chemical calculations of the formation free energies of selected MA-MSA-OA ternary clusters and on results obtained from ACDC (Atmospheric Cluster Dynamics Code) simulations. The main conclusion of the paper is that the formic acid (ForA) molecule is of particular interest because ForA might have an important role in MSA-driven new particle formation (NPF) in relevant atmospheric conditions. The various factors that affect the enhancing potential of the organic acids, especially ForA, on MSA-MA NPF were also thoroughly analysed.*

*The work is technically well performed and the results of the calculations well support the conclusions. The manuscript is almost clearly written and should be interesting for the community of atmospheric chemists.*

*I recommend publication of this manuscript after the following (minor) points have been taken into account.*

**AC:** Thanks for the positive comments. We have revised the manuscript to further enhance its quality.

**Special Suggestions and Comments**

**RC:** 1) *When comparing the various dicarboxylic acid molecules considered in the calculations, it is found that only maleic and glutaric acid can interact via their two carboxylic groups. It is however suprising that such configuration has not been found for Succinic acid. This has to be discussed.*

**AC:** Thanks for the suggestion. We agree with the reviewer that we should discuss why other dicarboxylic acids interact with $(MA)_1(MSA)_1$ only via one -COOH group. In the revised manuscript, the following sentences have been added:

   "For OxaA, the configuration involving the interaction of two -COOH groups with $(MA)_1(MSA)_1$ was not located. This results from the fact that the two -COOH groups in OxaA are directly linked and thereby OxaA cannot adapt a conformation where both -COOH groups interact with other molecules in the clusters. For MalA, SucA and AdiA, the binding free energies for the configuration with interactions of both -COOH groups with $(MA)_1(MSA)_1$ are higher than that for the configuration with the interaction of single -COOH group with $(MA)_1(MSA)_1$. This could result from the high deformation energy penalty when these three OAs interact with $(MA)_1(MSA)_1$ via two -COOH groups. An interesting phenomenon was that MaleA and SucA have different interaction pattern although they have the same number of C-atom. Such difference mainly results from the existence of double bond linker between the two -COOH groups of MaleA. The double bond linker allows MaleA to interact with $(MA)_1(MSA)_1$ via two -COOH groups and slight deformation as shown in Fig. S1."

We added the Figure S1 in the revised Supplement.

[Figure]

**MaleA**   **(MSA)₁(MA)₁(MaleA)₁**

**Figure S1. Lowest Gibbs free energy conformations of MaleA and (MSA)₁(MA)₁(MaleA)₁ cluster at the ωB97X-D/6-31++G(d,p) level of theory. The red balls represent oxygen atoms, blue ones for nitrogen atoms, gray ones for carbon atoms, and white ones for hydrogen atoms.**

**RC:** 2) *Although the ACDC code has been presented elsewhere, it is quite disappointing not having here a brief presentation of its main inputs and outputs. In particular, this would greatly improve understanding of section 3.4 and Figure 6.*

**AC:** Thanks for the comment. We agree that a brief presentation of basic formula and main inputs and outputs of ACDC should be provided. In the revised manuscript, the following sentences have been added to present basic formula and main inputs and outputs of ACDC:

"Briefly, the core of ACDC is to employ the birth-death equation (Eq. (2)) to describe the time-dependent cluster distributions:

$$\frac{dc_i}{dt} = \frac{1}{2}\sum_{j<i}\beta_{j,(i-j)}c_j c_{(i-j)} + \sum_j \gamma_{(i+j)\rightarrow i}c_{i+j} - \sum_j \beta_{i,j}c_i c_j - \frac{1}{2}\sum_{j<i}\gamma_{i\rightarrow j}c_i + Q_i - S_i \tag{2}$$

where subscripts ($i, j, i\text{-}j, j\text{-}i$ and $i+j$) represent different clusters or monomers in the system, $c_i$ represents the number concentration of $i$, $\beta_{i,j}$ denotes the collision rate coefficient between $i$ and $j$, $\gamma_{(i+j)\rightarrow i}$ denotes the evaporation rate of a cluster $i+j$ into smaller clusters (or monomer) $i$ and $j$. $Q_i$ represents an additional outside source term of $i$ and $S_i$ represents other sink terms for $i$. The collision rate coefficients were calculated by hard sphere kinetic gas theory as:

$$\beta_{i,j} = \left(\frac{3}{4\pi}\right)^{\frac{1}{6}}\left(\frac{6k_bT}{m_i} + \frac{6k_bT}{m_j}\right)^{\frac{1}{2}}\left(V_i^{\frac{1}{3}} + V_j^{\frac{1}{3}}\right)^2 \tag{3}$$

where $k_b$ is the Boltzmann constant, $T$ is the temperature, and $m_i$ and $V_i$ are the mass and volume of $i$, respectively. The evaporation rates were calculated using detailed balance as:

$$\gamma_{(i+j)\rightarrow i} = \beta_{i,j}c_{\text{ref}}\exp\left\{\frac{\Delta G_{i+j} - \Delta G_i - \Delta G_j}{k_bT}\right\} \tag{4}$$

where $\Delta G$ is the formation free energy of the cluster, $c_{\text{ref}}$ is the reference monomer concentration at 1 atm, which is the pressure at which $\Delta G$ was calculated (Mcgrath et al., 2012)."

"We used the calculated thermodynamic data of (MSA)$_x$(MA)$_y$(OA)$_z$ ($0 \leq y \leq x+z \leq 3$) clusters as input for ACDC simulations to obtain the cluster formation pathways and new particle formation rates."

**RC:** 3) *Finally, when comparing the DeltaG values, it should be clearly stated that the discussion in the paper is based on the absolute values, whereas relative values are given in Tables and Figures. Then, « higher values » in the text correspond in fact to « lower values » in the Tables/Figures.*

**AC:** We are actually already discussing the relative values in the text. For instance, at Line 145 we write: "Generally, the $\Delta G$ values of the $(MSA)_1(MA)_1(OA)_1$ clusters vary from -12.69 to -17.87 kcal mol$^{-1}$, and are 2.49-7.67 kcal mol$^{-1}$ higher than that of the $(MSA)_2(MA)_1$ cluster." In this context higher refers to less negative, which is also the case if you look at Figure 2a (compare the gray bar of MSA with the brown bars of the OAs). However, we agree with the reviewer that this should be clearer in the text as it is always convoluted to discuss negative quantities. We have modified the first occurrence where we discuss the free energies (i.e. the above sentence) to reflect this issue. In the revised manuscript, the original sentence "Generally, the $\Delta G$ values of the $(MSA)_1(MA)_1(OA)_1$ clusters vary from -12.69 to -17.87 kcal mol$^{-1}$, and are 2.49-7.67 kcal mol$^{-1}$ higher than that of the $(MSA)_2(MA)_1$ cluster." was revised as

"Generally, the $\Delta G$ values of the $(MSA)_1(MA)_1(OA)_1$ clusters vary from -12.69 to -17.87 kcal mol$^{-1}$, and are 2.49-7.67 kcal mol$^{-1}$ higher (i.e. less negative) than that of the $(MSA)_2(MA)_1$ cluster."